# Reduction of Deoxynivalenol in Wheat with Superheated Steam and Its Effects on Wheat Quality

**DOI:** 10.3390/toxins11070414

**Published:** 2019-07-16

**Authors:** Yuanxiao Liu, Mengmeng Li, Ke Bian, Erqi Guan, Yuanfang Liu, Ying Lu

**Affiliations:** National Engineering Laboratory for Wheat and Corn Further Processing, Henan University of Technology, Zhengzhou 450001, China

**Keywords:** deoxynivalenol, wheat, superheated steam, wheat quality, crisp biscuit

## Abstract

Deoxynivalenol (DON) is the most commonly found mycotoxin in scabbed wheat. In order to reduce the DON concentration in scabbed wheat with superheated steam (SS) and explore the feasibility to use the processed wheat as crisp biscuit materials, wheat kernels were treated with SS to study the effects of SS processing on DON concentration and the quality of wheat. Furthermore, the wheat treated with SS were used to make crisp biscuits and the texture qualities of biscuits were measured. The results showed that DON in wheat kernels could be reduced by SS effectively. Besides, the reduction rate raised significantly with the increase of steam temperature and processing time and it was also affected significantly by steam velocity. The reduction rate in wheat kernels and wheat flour could reach 77.4% and 60.5% respectively. In addition, SS processing might lead to partial denaturation of protein and partial gelatinization of starch, thus affecting the rheological properties of dough and pasting properties of wheat flour. Furthermore, the qualities of crisp biscuits were improved at certain conditions of SS processing.

## 1. Introduction

Head blight is a kind of fungal disease affecting wheat plant throughout the world. Both the yield and the quality of wheat are seriously affected by this disease. What’s worse, *Fusarium* fungi in wheat grains can produce many kinds of mycotoxins, among which deoxynivalenol (DON) is the most commonly found [1]. In 2017, the DON levels in 48 wheat samples harvested in Brazil range from 1329 to 3937 μg/kg and the average level is 2398 μg/kg [2]. Survey of 54 wheat samples in Southeast Romania in 2009 indicated that 66% of the samples were positive with DON with the average level of 4772 μg/kg [3]. In 2014, a three-year survey of the occurrence of DON in wheat in some regions of Jiangsu Province, China indicated that DON was found in 74.4% of samples at levels ranging from 14.52 to 41,157.13 μg/kg [4]. Besides, a survey of the global occurrence of deoxynivalenol in food commodities and exposure risk assessment in humans in the last decade showed that the problem of DON contamination in wheat is becoming more and more terrible [5]. Since wheat grains are common materials of foods and feeds, DON in wheat may be toxic to humans and animals. High concentration of DON has acute toxic effects, such as vomiting and diarrhea, on humans and animals. Meanwhile, taking in DON for a long period may do harm to the immune system, hematopoietic system, stomach, bone marrow and lymphoid tissues [6]. Scientists all over the world have done much research into the degradation and reduction methods of DON, such as heat treatment [7], activated carbon adsorption [8], ozone fumigation [9], ray irradiation [10], alkali steeping [11] and biodegradation [12]. However, many of these methods are hard to apply in wheat processing up to now. Therefore, it is meaningful to explore new means for reducing DON in wheat and their application in food processing.

Superheated steam (SS) is a kind of steam with temperature above the saturation point of water at given pressure. Compared to hot air, SS has some advantages, including higher drying rate, oxygen-free environment, free of pollution and energy conservation [13]. For these features, SS has been used in the drying of wood [14], paper [15], coal [16], sludge [17], etc. In recent years, researchers have tried to apply SS to the processing of food products. They mainly focus on the drying of foodstuffs, such as potato slices [18], oat groats [19], carrot [20], pork [21], beet, etc. What’s more, drying of beet with SS has been applied into some factories in the world [13]. Meanwhile, researchers have also studied the application of SS in food sterilization [22], toxin degradation [23] and enzymatic inactivation [24]. 

DON can be damaged when temperature is high enough and many studies on the stability of DON during different kinds of heat treatment have been carried out [25,26,27,28]. Therefore, SS may be an available way to reduce the content of DON in scabbed wheat. Besides, much of DON can be reduced when wheat kernels are milled into wheat flour. With the combination of wheat milling and SS treatment, a lot of DON in wheat flour may be reduced. Wheat protein may be partially denatured at high temperature, thus leading to the weakening of gluten strength. As a result, the scabbed wheat processed with SS can be milled into low-gluten wheat flour, which is fit to make crisp biscuits and cakes. The objectives of this study were to: (i) investigate the application of SS (with temperature above 185 °C) to reduce the content of DON in scabbed wheat; (ii) explore the effect of SS on wheat quality; (iii) understand the effect of SS on the quality of crisp biscuits. 

## 2. Results and Discussion

### 2.1. Reduction of Deoxynivalenol 

#### 2.1.1. Steam Temperature

In order to study the effect of steam temperature on DON reduction, the other two factors were set as follows: processing time, 6 min; steam velocity, 3 m/s. Both the concentration of DON in wheat kernels and wheat flour reduced significantly with the increase of steam temperature (Figure 1). This was because that the increase of temperature led to the stability weakening of DON. Since the surrounding environment of wheat kernels was oxygen-free during SS processing, oxidation could hardly occur. Therefore, large quantities of heat might be the primary factor leading to DON degradation. Charlene, Wolf and Wolf-Hall has drawn the similar conclusion [29,30,31]. In all the experiments, the reduction rate of DON in wheat kernels was always higher than that of wheat flour. The highest reduction rate of DON in wheat kernels was 77.4%, while it was only 60.5% for wheat flour. This phenomenon might due to the fact that DON was mainly distributed in the pericarp of wheat grain and heat was transferred from the outer surface to the inner endosperm. As a result, the DON in the endosperm couldn’t absorb as much heat as that in the bran of wheat. Since high temperature might do much harm to grain quality, the steam temperature used in practical application should be determined based on the real content of DON in wheat and the changes of wheat quality. 

#### 2.1.2. Processing Time

In order to study the effect of processing time on the reduction of DON, the other two factors were set as follows: steam temperature, 225 °C; steam velocity, 3 m/s. In the initial stage of SS processing, heated steam condensed on the cool surface of wheat kernels. During this stage, some DON on the surface of wheat kernels might be washed away as well as thermally degraded [7]. At different temperature, the quantity of condensed water was different. As a result, the time needed to evaporate the condensation was also different [32,33]. After the above stage, the temperature of wheat kernels began to rise and condensed water began to be evaporated. During this course, little heat was transferred into the inner of wheat kernels. As a result, when processed with SS for 2 min, some DON in the pericarp could be degraded, while little DON in wheat flour was reduced (Figure 2). When processing time was increased, considerable heat was transferred into the inner of wheat kernels. As a result, the concentration of DON in wheat and wheat flour decreased significantly and the reduction rate increased notably. Because of the harmful effect of SS on wheat quality, the processing time should be chosen according to the changes of DON content and wheat quality. In most experiments, the reduction of DON in wheat kernels was higher than that of wheat flour. However, when processing time was above 8 min, the gap was narrowed. This result suggested that when processing time was long enough, enough heat could be transferred into the endosperm of wheat. 

#### 2.1.3. Steam Velocity

In order to study the effect of steam velocity on the reduction of DON, the other two factors were set as follows: steam temperature, 225 °C; processing time, 6 min. Steam velocity had great effects on the heat transfer efficiency of SS, thus affecting the reduction of DON significantly. Different variation trends of DON concentration in wheat kernels and wheat flour were observed when steam velocity was increased. For wheat kernels, when steam velocity was increased from 1 m/s to 3 m/s, the concentration of DON in wheat kernels decreased significantly (Figure 3); when steam velocity was increased from 3 m/s to 5 m/s, the DON concentration increased. However, for wheat flour, DON concentration decreased significantly when steam velocity was increased from 1 m/s to 5 m/s. These results indicated that high velocity of SS might enhance the transfer of heat from the surface of wheat to the inner endosperm. However, when steam velocity was increased to 5 m/s, the reduction rate of DON in wheat flour was much higher than that of wheat kernels. Since more energy will be consumed when steam velocity was increased, both energy conservation and DON content should be taken into consideration when choosing the optimum steam velocity. 

The above results showed that steam temperature, processing time and steam velocity all had significant influence on the reduction of DON. Meanwhile, wheat quality may also be affected negatively. Therefore, under the condition that safety is insured, lower steam temperature, less time and suitable steam velocity should be used during SS processing. Compared to the research by Pronky [7], the reduction rate of DON in wheat was increased from 52% to 77.4% by improving the conditions of SS processing in this research. Besides, SS processing might be a more effective method to reduce DON compared to other methods of treatment, such as cooking, baking and extrusion [25,26,27,28]. However, no instrument exists now for the processing of wheat with SS in the industry. 

### 2.2. Rheological Properties of Dough

Wheat flour milled with Sample L was used to study the effects of SS processing on rheological properties of dough. Rheological properties of dough are closely related to the quality of food made with wheat flour. Rheological properties obtained from Chopin Mixolab are positively correlated with those obtained from Brabender Farinograph [34]. Therefore, rheological qualities determined by Chopin Mixolab can reflect the quality of wheat flour very well. Water absorption increased significantly with the increase of steam temperature, processing time and steam velocity (Table 1). This might due to the increased content of damaged starch in wheat caused by heat during SS processing [35]. For all the samples, the development time and stability decreased significantly when wheat kernels were processed with SS. This phenomenon resulted from the denaturation of gliadin and glutenin in wheat at high temperature, which made them unable to form gluten network with enough strength. Besides, SS processing also led to the aggregation of gluten proteins, which decreased the strength of dough obviously [36]. Sulfydryl and disulfide bond in wheat protein are the two most important groups for the formation of gluten matrix. However, they were easy to be changed during SS processing. Firstly, SS treatment led to the reduction of free sulfydryl and the cross-linking of glutenin [37]. Besides, SS processing affected the hydratability of wheat protein, thus changing the water absorption and the development of dough [38]. In a word, the destructive effect of SS processing on wheat protein led to the shortening of development time and stability time of dough. As a result, the qualities of crisp biscuits might be affected. The detailed contents will be discussed latterly. 

### 2.3. Pasting Properties of Wheat Flour

Wheat flour milled with Sample L was used to study the effects of SS processing on pasting properties of wheat flour. Starch, which accounts for about 75%–80% of wheat endosperm, is the most abundant component in wheat flour. Pasting properties can reflect the characteristics of starch in wheat flour. The SS treatment of wheat kernels might lead to starch gelatinization, retrogradation and the generation of damaged starch, thus changing the pasting properties of wheat flour. Analyzing Table 2, the following results could be found. All the pasting properties of flour treated with SS were higher than those of the control sample. With the increase of steam temperature, peak viscosity (PV), hold viscosity (HV), breakdown (BD), final viscosity (FV), setback (SB) and time to peak (TP) showed trends of decreasing. However, pasting temperature (PT) didn’t change significantly. When processing time was increased from 2 min to 10 min, PV, HV, BD, FV, SB and TP showed trends of decreasing while PT didn’t change significantly. Besides, the PT at every processing time was higher than that of the untreated sample. With the increase of steam velocity all the parameters showed trends of decreasing except PT while PT didn’t change significantly. 

The changes of pasting properties mainly resulted from the gelatinization and retrogradation of wheat starch and damaged starch caused by SS treatment [35,39]. Besides, the hydrophobicity of wheat albumin, globulin and starch were modified during SS processing [40,41]. The above changes might finally increase the water absorbing capacity of starch granules, which led to the increase of PV, TP and PT [42]. The damaged starch in wheat flour might lead to the decrease of PV, HV, BD, FV, SB and PT [43,44,45]. In addition, the activity of α-amylase were damaged for some degree during SS processing, thus bringing about the decrease of BD [46].

### 2.4. Quality of Crisp Biscuits

Crisp biscuits made with Sample L were used to study the effects of SS processing on the qualities of crisp biscuits. Although the qualities of wheat were affected by SS greatly, the wheat processed by SS was still suitable for making some kinds of food products, such as biscuits, cakes and pastries. In this study, crisp biscuits were chosen as the final products to study whether the wheats processed by SS were suitable for making food products or not.

The effects of steam temperature on the qualities of biscuits were as follows (Table 3 and Table 4). The hardness and working value of biscuits showed trends of decreasing with the increase of steam temperature, although they decreased non-significantly during some ranges. The above results indicated that the increase of steam temperature improved the crisping of crisp biscuits. However, when the temperature excessed 225 °C, the color of biscuit surfaces began to darken and the number of cracks increased. 

The effects of processing time on the qualities of biscuits were as follows (Table 3 and Table 5). The hardness and working value of biscuits showed trends of decreasing with the increase of processing time. However, when processing time was too long (≥8 min), the color of biscuit surfaces began to darken and the number of cracks increased. 

The effects of steam velocity on the qualities of biscuits were as follows (Table 3 and Table 6). The hardness and working value of biscuits showed trends of decreasing with the increase of steam velocity, although they decreased non-significantly during some ranges. However, when steam velocity reached 4 m/s, the color of biscuit surfaces began to darken and the number of cracks increased. 

The variations of biscuit quality were closely related to the changes of dough rheology and flour pasting properties. To some degree, the shortening of dough stability time made the wheat flour more suitable for making crisp biscuits. However, short time of stability was still needed. Besides, SS processing of wheat kernels at high temperature or for long time might result in the color change of wheat flour, which had negative effects on biscuit color. Therefore, suitable conditions for processing wheat kernels should be considered to avoid the deterioration of biscuit quality to the greatest extent. 

## 3. Conclusions

SS had proven to be an effective method to reduce the content of DON in wheat and wheat flour. DON on the surface of wheat kernels could be washed as well as thermally degraded, while DON in the inner of wheat kernels could be reduced by thermal degradation. With the increase of steam temperature and processing time, the reduction rate increased significantly. Stem velocity also had significant influence on the reduction rate. In this research, the reduction rate of DON could reach 77.4% for wheat kernels, while it was 60.4% for wheat flour. Besides reducing DON in wheat, superheated steam also had great effects on wheat quality, thus affecting the qualities of food made from wheat flour. During SS processing, protein denaturation and starch gelatinization occurred in wheat. The degree of protein denaturation and starch gelatinization depended on the conditions of SS processing. As a result, the water absorption of dough was increased, while the development time and stability were decreased. In addition, the pasting properties of wheat flour were affected significantly due to the increase of damaged starch, the gelatinization of starch and the passivation of α-amylase. What’s more, the qualities of crisp biscuits made with SS-processed wheat could be improved under certain conditions. Last but not least, if we want to use SS treatment in the industry, we just need to combine a steam generator with a heater to produce SS and the residual can be recycled by a centrifuge fan. The device is quite easy to design and manufacture. Therefore, for scabbed wheats, superheated steam processing could be an effect way to reduce DON as well as improving the qualities of crisp biscuits made from processed wheat and it is likely to be used in practical production. In further research, we will aim at the analysis of DON degradation products and the toxicity of degradation products. Besides, the mechanism of the quality change of wheat will also be studied further. 

## 4. Materials and Methods

### 4.1. Materials and Reagents 

Scabbed wheat with about 3.8 mg/kg of DON (Sample H) was harvested on a farmland of Xinyang City, Henan Province. Scabbed wheat with about 2.3 mg/kg of DON (Sample L) was harvested on a farmland of Nanyang City, Henan Province. All samples were cleaned and stored at 10 °C and no more than 50% of relative humidity before experiments. 

### 4.2. Determination of Moisture Content

Moisture content of wheat and wheat flour were tested according to AACC (American Association of Cereal Chemists) Method 44-15A.

### 4.3. Tempering of Wheat

Tempering of wheat was conducted according to AACC Method 26-10A.

### 4.4. Superheated Steam Processing

The power was connected and the device was turned on in advance. When beginning the experiment, 200 g of wheat kernels were placed on the screen mesh of a sieve with 1.98 mm of aperture. Steam temperature was changed from 185 °C to 265 °C (185 °C, 205 °C, 225 °C, 245 °C and 265 °C) to study the effect of steam temperature on experimental results. Processing time was changed from 2 min to 10 min (2 min, 4 min, 6 min, 8 min and 10 min) to study the effect of processing time on experimental results. Steam velocity was changed from 1 m/s to 5 m/s (1 m/s, 2 m/s, 3m/s, 4 m/s and 5 m/s) to study the effect of steam velocity on experimental results. Put the screen mesh with wheat kernels into the processing chamber of SS device (Figure 4). During processing, steam passed through the wheat kernels in the processing chamber. At the beginning of processing, steam condensed on the surface of wheat kernels and then the condensed water and the water in wheat kernels was evaporated. When the processing procedure was finished, wheat kernels were moved out of the processing chamber as quickly as possible and weighed immediately. For each condition, about 3000 g of sample is needed which equaled to 15 times of SS processing. The processed wheat kernels were cooled at room temperature on the screen mesh. After cooling, mix the samples processed at the same condition. The processed samples were moved into airtight bags and stored at room temperature for no more than 24 h before smashing and milling. Then, 100g of processed wheat kernels were smashed (4.5) to determine the moisture and DON concentration and others were milled into wheat flour (4.6).

### 4.5. Wheat Smashing

200 g of wheat kernels were smashed in the high-speed universal grinder for about 45 s. Fractured samples were sifted with a sieve with mesh number of 40 and the undersized parts were collected. The oversized parts were put in the grinder and smashed for 20 s again. Sift the fractured materials with a sieve with mesh number of 40 and collect the undersized parts. Then the undersized parts were mixed and stored at room temperature.

### 4.6. Wheat Milling

Wheat kernels were milled according to AACC Method 26–31 and Bühler model MLU-202 was used as the milling instrument. The feed rate was adjusted to 80 g/min. The sifters of the flour mill were cleaned thoroughly between of milling of two different samples. After milling, the wheat flour was moved into airtight bags and stored in the refrigerator at 4 °C. 

### 4.7. Extraction and Purification of DON

The extraction and purification of DON were conducted according to the method as follows [9].

Extraction of DON: Weigh 25 g of smashed wheat (or 20 g of wheat flour) in a beaker. 100 mL of extracting solution (*V_acetonitrile_/V_water_* = 84/16) was added into the beaker. After then, the sample and solution were stirred with a magnetic stirrer for 20 min. 

Purification of DON: After the above steps, keep the beaker still for at least 5 min, and then 5 mL of the supernatant was transferred into a SPE (Solid Phase Extraction) column to wipe off some of the impurities in the DON solution. 4 mL of the purified solution was transferred into a tube and dried with nitrogen. The residue was re-dissolved with 2 mL of mobile phase (*V_acetonitrile_/V_water_ =* 16/84). All the re-dissolved solution was transferred into a centrifuge tube and centrifuged at the speed of 12,000 r/min. Then, the supernatant was transferred into a vial through a filter membrane with 0.22 μm of pore size. All the samples were stored at 4 °C in a dark environment. 

### 4.8. Preparation of Standard Solution of DON

Stock solution of DON (100 mg/L, 1 mL) was transferred into a test tube (10 mL). The solution was dried with nitrogen gas and the remained DON was re-dissolved with mobile phase (*V_acetonitrile_/V_water_ =* 16/84). The DON solution was diluted successively to get DON solution with the following concentration: 0.1 mg/L, 0.2 mg/L, 0.5 mg/L, 1 mg/L, 2 mg/L, 5 mg/L and 10 mg/L.

### 4.9. HPLC Analysis

DON contents were determined by high performance liquid chromatography (HPLC) with Symmetry C18 column (250 mm × 4.6 mm i.d., 5-μm particle size) and ultraviolet (UV) detector. The HPLC conditions were optimized based on the method of Cui [47]. The mobile phase was prepared from acetonitrile and water (*V*_acetonitrile_/*V*_water_= 16/84). The flow rate was 600 μL/min. The absorbance of DON at 218 nm was monitored and the full running time was 10 min. The retention time (RT) of DON was about 7.6 min and the limit of quantification (LOD) for DON was 0.03 μg/mL.

### 4.10. Determination of Dough Rheological Properties 

Rheological properties were analyzed with Chopin Mixolab and AACC Method 54–60 was used as reference.

### 4.11. Analysis of Pasting Properties 

Pasting properties were determined with Rapid Viscosity Analyzer (RVA) according to AACC Method 76-21.

### 4.12. Manufacturing of Crisp Biscuits

The method of making crisp biscuits was modified according to LS/T 3206-1993. The detailed steps were as follows: Powdered sugar (28.5 g), sodium bicarbonate (0.6 g), and salt (0.3 g) were mixed in 5 mL of water and stirred to accelerate dissolution. 33 g of shortening was melted by hot-water bath and added into the above solution. The above ingredients were stirred until they formed emulsion. 100 g of wheat flour was added into the emulsion. Then all the ingredients were mixed by dough mixer to accelerate the formation of dough. The dough was kept still for nearly 10 min until it was not sticky. After this, the dough was tableted until the thickness reached about 3.5 mm–4 mm. Then it was molded and baked for 10 min. 

### 4.13. Texture Analysis 

Texture properties of biscuits were determined with TA-XT2i texture analyzer (STABLE MICRO SYSTEM). The main parameters were set as follows: 5 mm/s of pre-test speed, 1 mm/s of test speed, 4 mm of target displacement and 5 g of trigger force.

### 4.14. Apparent Conditions of Biscuits

Crisp biscuits were taken photos of to observe the apparent conditions of crisp biscuits.

### 4.15. Statistical Analysis

Data are shown as the mean ± SD of at least three parallel experiments. One-way ANOVA (analyses of variance) approach was used to compare values among more than two different experimental groups. Duncan multiple comparison method was used and *p* values less than 0.05 were considered statistically significant. Statistics were analyzed with SPSS 16.0 (SPSS Institute, CHI, USA). All the diagrams were drawn with Origin 8.5 (OriginLab Corporation, MA, USA).

## Figures and Tables

**Figure 1 toxins-11-00414-f001:**
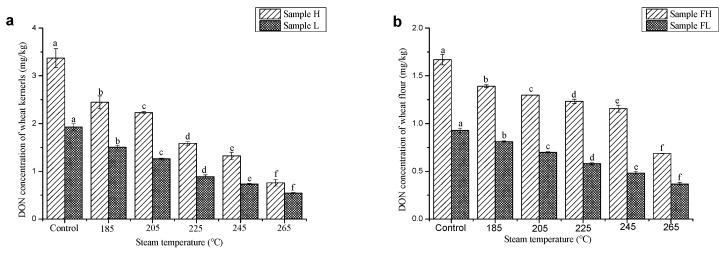
Effect of steam temperature on the contents of DON in wheat kernels (**a**) and wheat flour (**b**) * Sample H: Scabbed wheat sample with relatively higher content of DON; Sample L: Scabbed wheat sample with relatively lower content of DON; FH: Wheat flour milled from Sample H; FL: Wheat flour milled from Sample L. Same designations were used in Figure 2. and Figure 3. Processing time and steam velocity are fixed at 6 min and 3 m/s respectively. Data are given as means of triplicate assays ± SD (standard deviation). Values with different letters on the same kind of bars are significantly different (*p* < 0.05). The bars are labeled by the letters a to f from the highest to the lowest. The same expressing methods are used in Figure 2 and Figure 3.

**Figure 2 toxins-11-00414-f002:**
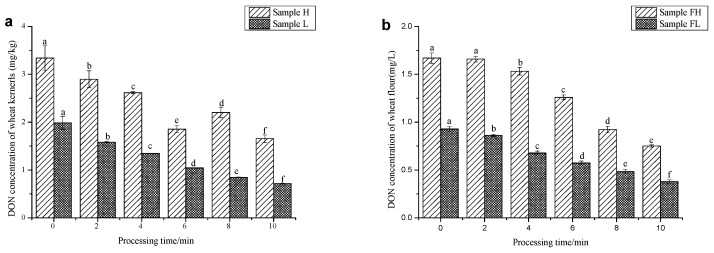
Effect of processing time on the contents of DON in wheat kernels (**a**) and wheat flour (**b**). * Steam temperature and steam velocity are fixed at 225 °C and 3 m/s respectively.

**Figure 3 toxins-11-00414-f003:**
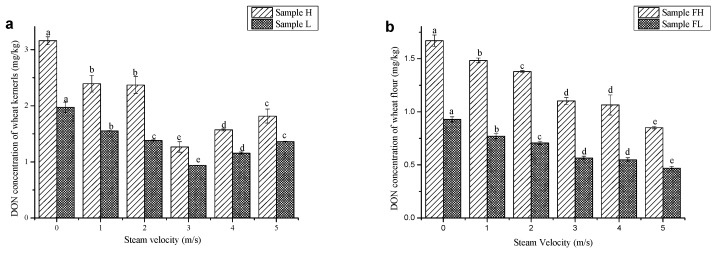
Effect of steam velocity on the contents of DON in wheat kernels (**a**) and wheat flour (**b**). * Steam temperature and processing time are fixed at 225 °C and 6 min respectively.

**Figure 4 toxins-11-00414-f004:**
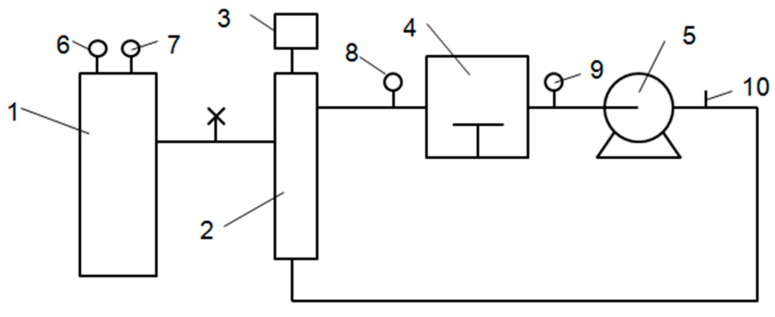
Schematic diagram of superheated steam device (1, Steam generator; 2, Heater; 3, Thermostat; 4, Processing chamber; 5, Centrifugal fan; 6, Pressure gauge; 7, 8, 9, Thermometer; 10, Flowmeter.).

**Table 1 toxins-11-00414-t001:** Effect of steam temperature on rheological properties of dough.

Processing Condition		Water Absorption/%	Development Time/min	Stability /min
	Control	57.5 ± 0.1 ^f^	3.01 ± 0.04 ^a^	6.29 ± 0.11 ^a^
Steam temperature (°C)	185	61.5 ± 0.1 ^e^	2.17 ± 0.06 ^b^	2.24 ± 0.11 ^b^
	205	65.2 ± 0.2 ^d^	2.06 ± 0.06 ^b^	2.18 ± 0.06 ^b^
	225	66.7 ± 0.1 ^c^	1.87 ± 0.03 ^c^	1.95 ± 0.08 ^c^
	245	71.9 ± 0.1 ^b^	1.90 ± 0.03 ^c^	1.86 ± 0.03 ^c^
	265	74.3 ± 0.1 ^a^	2.08 ± 0.04 ^b^	1.83 ± 0.03 ^c^
Processing time (min)	2	62.2 ± 0.1 ^e^	2.50 ± 0.04 ^b^	2.28 ± 0.04 ^b^
	4	65.5 ± 0.1 ^d^	2.10 ± 0.05 ^c^	2.16 ± 0.04 ^bc^
	6	68.1 ± 0.1 ^c^	1.94 ± 0.04 ^d^	2.15 ± 0.07 ^bc^
	8	70.4 ± 0.1 ^b^	2.01 ± 0.01 ^cd^	2.02 ± 0.01 ^c^
	10	72.3 ± 0.1 ^a^	2.03 ± 0.02 ^c^	2.01 ± 0.03 ^c^
Steam velocity (m/s)	1	62.2 ± 0.1 ^e^	2.21 ± 0.04 ^b^	2.02 ± 0.03 ^d^
	2	63.6 ± 0.2 ^d^	2.09 ± 0.03 ^c^	2.18 ± 0.08 ^c^
	3	67.7 ± 0.1 ^c^	1.95 ± 0.04 ^d^	2.30 ± 0.04 ^bc^
	4	70.7 ± 0.1 ^b^	2.08 ± 0.08 ^c^	2.33 ± 0.02 ^bc^
	5	75.6 ± 0.2 ^a^	2.25 ± 0.03 ^b^	2.38 ± 0.04 ^b^

Data are given as means of triplicate assays ± SD. The values are labeled by the letters a to f from the maximum to the minimum. Values with different letters in the same column are significantly different (*p* < 0.05).

**Table 2 toxins-11-00414-t002:** Effect of steam temperature on pasting properties of wheat flour.

Processing Condition		PV/mPa·s	HV/mPa·s	BD/mPa·s	FV/mPa·s	SB/mPa·s	TP/min	PT/ °C
	Control	2707 ± 11 ^c^	1797 ± 23 ^d^	910 ± 23 ^a^	3026 ± 34 ^c^	1229 ± 11 ^b^	6.34 ± 0.09 ^a^	67.7 ± 0.1 ^b^
Steam temperature (°C)	185	3281 ± 64 ^a^	2390 ± 69 ^a^	891 ± 4 ^a^	3874 ± 49 ^a^	1484 ± 20 ^a^	6.20 ± 0.10 ^ab^	84.4 ± 0.6 ^a^
	205	3201 ± 13 ^ab^	2284 ± 30 ^ab^	917 ± 17 ^a^	3857 ± 21 ^a^	1574 ± 9 ^a^	6.13 ± 0.02 ^bc^	84.4 ± 0.5 ^a^
	225	3032 ± 41 ^b^	2247 ± 17 ^b^	785 ± 25 ^b^	3818 ± 13 ^a^	1571 ± 30 ^a^	6.00 ± 0.10 ^bc^	84.0 ± 1.2 ^a^
	245	2835 ± 112 ^c^	2203 ± 48 ^b^	632 ± 64 ^c^	3663 ± 122 ^a^	1460 ± 74 ^a^	5.97 ± 0.05 ^c^	83.6 ± 0.6 ^a^
	265	2486 ± 134 ^d^	1974 ± 81 ^c^	512 ± 53 ^d^	3275 ± 165 ^b^	1301 ± 84 ^b^	5.94 ± 0.09 ^c^	84.0 ± 0.1 ^a^
Processing time (min)	2	3214 ± 2 ^a^	2339 ± 21 ^a^	875 ± 23 ^ab^	3880 ± 4 ^a^	1541 ± 24 ^a^	6.17 ± 0.05 ^ab^	83.9 ± 1.2 ^a^
	4	3217 ± 26 ^a^	2377 ± 18 ^a^	840 ± 8 ^b^	3924 ± 43 ^a^	1547 ± 25 ^a^	6.17 ± 0.05 ^ab^	84.4 ± 0.5 ^a^
	6	2961 ± 30 ^b^	2244 ± 9 ^b^	717 ± 40 ^c^	3652 ± 88 ^b^	1408 ± 79 ^b^	6.07 ± 0.09 ^b^	83.9 ± 1.2 ^a^
	8	2567 ± 21 ^d^	2005 ± 23 ^c^	563 ± 2 ^d^	3414 ± 25 ^c^	1410 ± 49 ^b^	5.80 ± 0.10 ^c^	83.5 ± 0.5 ^a^
	10	1944 ± 10^e^	1533 ± 1 ^e^	412 ± 11 ^e^	2576 ± 4 ^e^	1044 ± 5 ^d^	5.57 ± 0.05 ^d^	84.0 ± 0.1 ^a^
Steam velocity (m/s)	1	3255 ± 38 ^a^	2345 ± 17 ^a^	910 ± 34 ^a^	3842 ± 64 ^a^	1497 ± 62 ^a^	6.11 ± 0.03 ^b^	84.2 ± 0.5 ^a^
	2	3162 ± 12 ^b^	2286 ± 34 ^ab^	876 ± 40 ^a^	3811 ± 72 ^ab^	1526 ± 97 ^a^	6.09 ± 0.08 ^bc^	84.2 ± 0.9 ^a^
	3	2966 ± 6 ^c^	2225 ± 51 ^b^	741 ± 57 ^b^	3723 ± 26 ^b^	1498 ± 77 ^a^	6.07 ± 0.09 ^bc^	84.4 ± 0.5 ^a^
	4	2707 ± 17 ^d^	2115 ± 31 ^c^	592 ± 52 ^c^	3506 ± 3 ^c^	1391 ± 32 ^ab^	5.97 ± 0.14 ^bc^	84.4 ± 0.6 ^a^
	5	2493 ± 23 ^e^	2008 ± 10 ^d^	485 ± 13 ^d^	3282 ± 10 ^d^	1274 ± 1 ^b^	5.90 ± 0.04 ^c^	84.0 ± 0.1 ^a^

PV, peak viscosity: the highest viscosity of sample after heating up and before cooling down; HV, hold viscosity: the minimum viscosity of sample during cooling down; BD, breakdown: the value of PV minus HV; FV, final viscosity: the viscosity of sample at the ending of test; SB, setback: the value of FV minus HV; TP, time to peak: the time when the viscosity reaches PV; PT, pasting temperature: the time when the viscosity of sample begins to increase. Data are given as means of triplicate assays ± SD. The values are labeled by the letters a to f from the maximum to the minimum. Values with different letters in the same column are significantly different (*p* < 0.05).

**Table 3 toxins-11-00414-t003:** Effect of steam temperature on texture properties of biscuits.

Processing Condition		Hardness/g	Working Value /g·s
	Control	2101 ± 186 ^a^	3896 ± 152 ^a^
Steam temperature (°C)	185	2093 ± 90 ^a^	3859 ± 22 ^a^
	205	1823 ± 95 ^b^	1942 ± 253 ^b^
	225	1552 ± 94 ^c^	1516 ± 117 ^bc^
	245	1466 ± 112 ^cd^	1213 ± 88 ^c^
	265	1253 ± 78 ^d^	1175 ± 69 ^c^
Processing time (min)	2	1708 ± 26 ^b^	2657 ± 178 ^b^
	4	1563 ± 38 ^bc^	2325 ± 60 ^bc^
	6	1396 ± 105 ^cd^	1750 ± 20 ^cd^
	8	1295 ± 98 ^d^	1623 ± 14 ^d^
	10	1497 ± 87 ^bcd^	1425 ± 30 ^d^
Steam velocity (m/s)	1	1898 ± 166 ^a^	2556 ± 129 ^b^
	2	1879 ± 3 ^ab^	2009 ± 111b ^cd^
	3	1656 ± 68 ^bc^	2377 ± 6 ^bc^
	4	1482 ± 76 ^cd^	1744 ± 1 ^cd^
	5	1254 ± 103 ^d^	1505 ± 6 ^d^

Data are given as means of triplicate assays ± SD. The values are labeled by the letters a to f from the maximum to the minimum. Values with different letters in the same column are significantly different (*p* < 0.05).

**Table 4 toxins-11-00414-t004:** Effect of steam temperature on apparent condition of biscuits.

Steam Temperature/°C	Control	185	205	225	245	265
Apparent condition	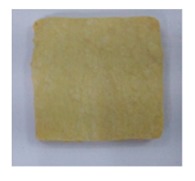	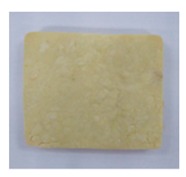	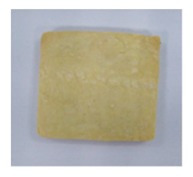	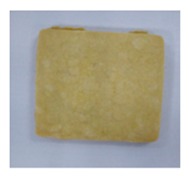	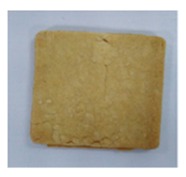	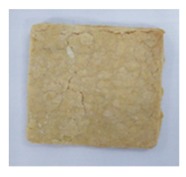

**Table 5 toxins-11-00414-t005:** Effect of processing time on apparent condition of biscuits.

Processing Time/min	Control	2	4	6	8	10
Apparent condition	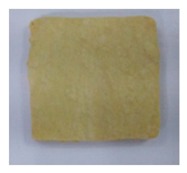	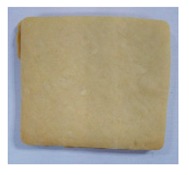	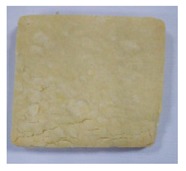	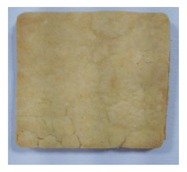	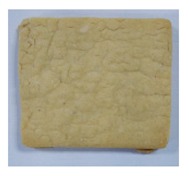	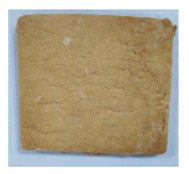

**Table 6 toxins-11-00414-t006:** Effect of steam velocity on apparent condition of biscuits.

Steam Velocity/(m/s)	Control	1	2	3	4	5
Apparent condition	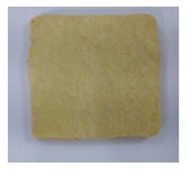	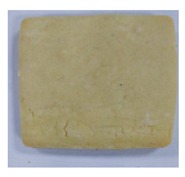	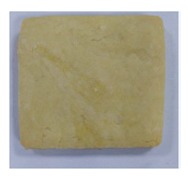	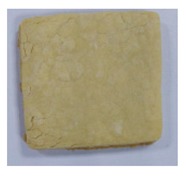	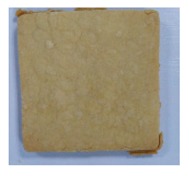	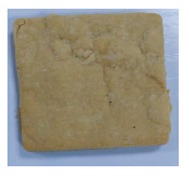

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
