# Peer review of "Reduction of Deoxynivalenol in Wheat with Superheated Steam and Its Effects on Wheat Quality"

_toxins, 2019, doi:10.3390/toxins11070414_

Round 1

Reviewer 1 Report

My biggest concern is - where did the DON go? Did it transfer into a different form, less or more toxic?

How did you get samples with such high DON concentrations? 

Does this marks the control samples "Scabbed wheat with about 3.8 mg/kg of DON (Sample H) was harvested on a farmland of 217 Xinyang City, Henan Province. Scabbed wheat with about 2.3 mg/kg of DON (Sample L)"?

This is nothing new, superheated steam is know to help reduce DON in wheat kernels for a great deal of time. I see no novelty or originality in this paper and therefore must suggest rejection. 

Author Response

Dear Reviewer:

Thank you very much for your suggestion and comments on our manuscript. Now I’m glad to reply to your comments point by point.

Point 1: My biggest concern is - where did the DON go? Did it transfer into a different form, less or more toxic?

Response 1: Thank you very much for your valuable comments on our paper.

As what you focus on our paper, we haven’t explained transformation way of DON during superheated steam treatment. We think what you say is quite important and we will list it in our research plan in the future. We will certainly do further research to know the transformation way and the toxicity of transformation products of DON and we believe we will get the results in the near future.

Point 2: How did you get samples with such high DON concentrations?

Response 2: The two samples are harvested on the farmlands in Henan Province. Because of the climate change and other factors, the problem wheat head blight is becoming more and more terrible all over the world, especially in China (please see the added references 2-5 at your convenience which were cited in Line 25-32). Therefore, the two samples can partially represent the current situation of wheat head blight in the world.

Point 3: Does this marks the control samples "Scabbed wheat with about 3.8 mg/kg of DON (Sample H) was harvested on a farmland of 217 Xinyang City, Henan Province. Scabbed wheat with about 2.3 mg/kg of DON (Sample L)"?

Response 3: The samples described in "Scabbed wheat with about 3.8 mg/kg of DON (Sample H) was harvested on a farmland of 217 Xinyang City, Henan Province. Scabbed wheat with about 2.3 mg/kg of DON (Sample L) was harvested on a farmland of Nanyang City, Henan Province " are control samples. By choosing the 2 samples, we want to know if the reduction rates of DON are significantly different if the starting levels of DON in wheat are different.

Point 4: This is nothing new, superheated steam is know to help reduce DON in wheat kernels for a great deal of time. I see no novelty or originality in this paper and therefore must suggest rejection.

Response 4: Although reduction of DON in wheat kernels by superheated steam has been known viable for a long time. However, the previous researches on the reduction of DON in wheat didn’t cover the DON concentration in wheat flour, rheological properties of dough or qualities biscuits, which were all included in our paper. Besides, the conditions we used for superheated steam treatment was different from the previous. In our research, we aimed to reduce the content of DON in wheat and tried to use the processed wheat as biscuit material at the same time and we have got some useful results. Besides, the materials in the previous researches were hard wheats, which were different from ours.

In conclusion, the previous researches has established a good basis for us and we have tried to make some improvements about this method.

Thank you for your comments on our paper and we will try to improve it continuously.

Reviewer 2 Report

The experiments with superheated steam on wheat kernels were run at laboratory scale and the experimental milling to produce flour does not represent a real situation in milling industry. No information is provided on the technological impact of superheated steam in commercial milling at industrial (large) scale. It is likely that the different milling fractions would be negatively affected as well as the performance of grinders, mills, rollers, sieves and most milling units.

Additionally, the level of contamination of the starting wheat kernels does not represent an admissible raw material for processing (milling) for human consumption. As a reference, EU maximum content for DON in unprocessed wheat is 1.25 mg/kg (1.75 mg/kg for durum wheat). The detoxification of foodstuffs not complying with certain safety levels, even with physical treatments such as heat, it is questionable for food protection reasons.

The results obtained (percentage of DON reduction) are not new as compared to those obtained by Pronyk et al. (Food Control 2006), who already described the reduction of >50% DON in wheat by superheated steam.

Introduction

Line 46: Why low-gluten wheat flour? The heat usually do not reduce the toxicity of gluten for celiac people.

Lines 56 and 70: Why does the numbering of the Figures begin with Figure 2? Is Figure 1 missing? (The same happens to Figures 3 and 4, they should be Figures 2 and 3, respectively)

Figure 2: What is the meaning of CK?

CK is used throughout manuscript and is not defined

Material and methods

4.1. What type of wheat was used, soft wheat or hard wheat?

4.5. What wheat smashing was conducted for? I do not understand why wheat smashing is described in detail while more important wheat milling (see 4.6)  is not described at all (only AACC reference is provided).

4.6. The description of milling is very important here. The quoted references are not enough. AACC Method 22-30A is not listed in AACC Methods list. For AACC Method 26-21A (useful for hard wheat) a Bühler laboratory mill is needed. Was Batch method used? What was the extraction percentage of the flour obtained? Depending of the method used for experimental milling the quality of flour varies.

Line 253: DON purification, what type of SPE column was used for cleanup?

Author Response

Dear Reviewer:

Thank you very much for your suggestion and comments on our manuscript. Now I’m glad to reply to your comments point by point.

Point 1: The experiments with superheated steam on wheat kernels were run at laboratory scale and the experimental milling to produce flour does not represent a real situation in milling industry. No information is provided on the technological impact of superheated steam in commercial milling at industrial (large) scale. It is likely that the different milling fractions would be negatively affected as well as the performance of grinders, mills, rollers, sieves and most milling units.

Response1: Thank you very much for your valuable comments on our paper.

In the results, we have supplemented some sentences to analyze the practicability of SS in milling industry (Line 247-249). Besides, what we plan do next is to do pilot plant test, thus establishing basis for the practical application of SS in the reduction of DON.

Point 2: Additionally, the level of contamination of the starting wheat kernels does not represent an admissible raw material for processing (milling) for human consumption. As a reference, EU maximum content for DON in unprocessed wheat is 1.25 mg/kg (1.75 mg/kg for durum wheat). The detoxification of foodstuffs not complying with certain safety levels, even with physical treatments such as heat, it is questionable for food protection reasons.

Response 2: Since the occurrence of wheat head blight is becoming more and more frequent and terrible, more wheat kernels contaminated with high concentration of DON are harvested every year in the world (please see the added references 2-5 at your convenience which were cited in Line 25-32). The purpose of our research is to try to reduce the contents of DON in these wheats to the contents lower than the maximum content stipulated by the government. For example, the DON concentration in wheat can be reduced to the level under the concentration stipulated by the government (1000 μg/kg). Therefore, I think the materials we use, especially sample L, can represent the current situation in China. Besides, I have also listed 4 more references (2-5, Line 25-32) to describe the current situation of DON contamination in the world, which shows that the starting level of DON we use are included in the most commonly found scabbed wheats. As for sample H, we want to use it to know the differences when superheated steam is used for reducing different starting contents of DON. At some conditions in our research, the level of DON in wheat and wheat flour can reach the safe level stipulated by the standards.

Point 3: The results obtained (percentage of DON reduction) are not new as compared to those obtained by Pronyk et al. (Food Control 2006), who already described the reduction of >50% DON in wheat by superheated steam.

Response 3: Compared to the research by Pronyk et al. (Food Control 2006), we have used different conditions of SS processing. As a result, we have obtained higher reduction rate (77.4%). Besides, the materials we use are soft wheats, which were different from Pronky’s (hard wheat) and represented the situation in China. What’s more, the previous research didn’t cover the DON concentration in wheat flour, rheological properties of dough or qualities biscuits, which were all important contents in our paper. The research we have done is on the basis of that by Pronky et al. and we have also made some progress.

Point 4: Line 46: Why low-gluten wheat flour? The heat usually do not reduce the toxicity of gluten for celiac people.

Response 4: As you say, superheated steam treatment do not reduce the toxicity for celiac people. However, what we want to express is that superheated steam treatment can reduce the gluten strength of wheat except for reducing DON concentration. As a result, the processed wheat can be used as the materials of crisp biscuits and we have done research into this topic in the following parts of our paper.

Point 5: Lines 56 and 70: Why does the numbering of the Figures begin with Figure 2? Is Figure 1 missing? (The same happens to Figures 3 and 4, they should be Figures 2 and 3, respectively)

Response 5: No figure is missing. I’m very sorry that the figure order has been arranged incorrectly by me and I have corrected it in the revised paper.

Point 6: Figure 2: What is the meaning of CK?

CK is used throughout manuscript and is not defined

Response 6: CK is the control sample. Since the word “CK” is not commonly used, I have replaced it with the word “control” in the revised paper.

Point 7: 4.1. What type of wheat was used, soft wheat or hard wheat?

Response 7: Since the occurrence area of Fusarium Head Blight in China mainly located between Yantze and Huai Rivers, which is the main producing area soft wheat, the two samples we use were all soft wheats.

Point 8: 4.5. What wheat smashing was conducted for? I do not understand why wheat smashing is described in detail while more important wheat milling (see 4.6) is not described at all (only AACC reference is provided).

Response 8: Wheat smashing was conducted for the extraction of DON. Since on standard method can be taken as reference, we must describe the smashing procedure in detail to make readers know it well. Besides, the procedure of wheat milling has also been described in the revised paper (Line 300-303).

Point 9: 4.6. The description of milling is very important here. The quoted references are not enough. AACC Method 22-30A is not listed in AACC Methods list. For AACC Method 26-21A (useful for hard wheat) a Bühler laboratory mill is needed. Was Batch method used? What was the extraction percentage of the flour obtained? Depending of the method used for experimental milling the quality of flour varies.

Response 9: I’m sorry that the AACC Method has been cited incorrectly. I’ve corrected it in the revised paper. The flour yields of all my control samples and treated samples are between 67% and 72%.

Point 10: Line 253: DON purification, what type of SPE column was used for cleanup?

Response 10: The type of SPE column we use is Bond Elut Mycotoxin SPE Column, which is suitable for the purification of trichothecenes.

Thank you for your comments on our paper and we will try to improve it continuously.

Reviewer 3 Report

The authors present the manuscript with the title: "Reduction of deoxynivalenol in wheat with 2 superheated steam and its effects on wheat quality." The work is very interesting, but I have some comments:

In the introduction, mention the reason why wheat, flour and biscuits were chosen from the point of view of DON concentrations. Mention from some recent reviews concerning the occurrence of DON what are the most frequent concentrations or some ranges for DON concentrations in these products in various countries in the world. It is important to mention this to understand if this process of SS will be able to have the expected results at the most frequent concentrations.  

Figure 2: A figure should be self-readable, thus please mention what is the difference between the graph a and graph b, and also mention what is the significance of the letters in the graphs.

Same comment also for Figure 3, 4 . 

Table 1: A table should be self-readable, thus please mention in the footer of the table what CK means, and also mention what is the significance of the letters in the table. Same comments also for Table 2 and Table 3. 

Reorganize information in Table 4, 5 and 6 making only one table. 

Line 219: I recommend mention also the humidity in the environment where the samples were stored. 

Regarding the HPLC method, from my point of view, it is necessary mention the limit of detection of the method and the limit of quantification for DON. 

In the Conclusion section, try to introduce some information about the applicability of the method. To have a valuable article, it is very important to underline how this information can help other future researches. 

Author Response

Dear Reviewer:

Thank you very much for your suggestion and comments on our manuscript. Now I’m glad to reply to your comments point by point.

The authors present the manuscript with the title: "Reduction of deoxynivalenol in wheat with superheated steam and its effects on wheat quality." The work is very interesting, but I have some comments:

Points 1: In the introduction, mention the reason why wheat, flour and biscuits were chosen from the point of view of DON concentrations. Mention from some recent reviews concerning the occurrence of DON what are the most frequent concentrations or some ranges for DON concentrations in these products in various countries in the world. It is important to mention this to understand if this process of SS will be able to have the expected results at the most frequent concentrations.

Responses 1: In the introduction, I have supplemented the reason why wheat, flour and biscuits were chosen from the point of view of DON concentrations (Line 54-56). Besides, 4 more references have been cited to describe the global occurrence of deoxynivalenol in wheat (Line 25-32). From the above 4 references, we can conclude that if this process of SS will be able to have the expected results at the most frequent concentrations.

Point 2: Figure 2: A figure should be self-readable, thus please mention what is the difference between the graph a and graph b, and also mention what is the significance of the letters in the graphs.

Same comment also for Figure 3, 4 .

Response 2: I’m sorry that I forgot to explain the meaning of a and b. The difference between a and b has been described in the figure caption in the revised paper (Line 81,110 ,141).

Point 3: Table 1: A table should be self-readable, thus please mention in the footer of the table what CK means, and also mention what is the significance of the letters in the table. Same comments also for Table 2 and Table 3.

Reorganize information in Table 4, 5 and 6 making only one table.

Response 3: CK is the control sample. Since the word “CK” is not commonly used, I have replaced it with the word “control” in the revised paper. The significance of the letters has been added in the revised paper (Line 165). I’m sorry that I haven’t made Table 4, 5 and 6 into one table because I found it hard to understand when making into one.

Point 4: Line 219: I recommend mention also the humidity in the environment where the samples were stored.

Response 4: Relative humidity has been added in the revised paper (Line 250).

Point 5: Regarding the HPLC method, from my point of view, it is necessary mention the limit of detection of the method and the limit of quantification for DON.

Response 5: The limit of quantification for DON has been added in the revised paper (Line329).

Point 6: In the Conclusion section, try to introduce some information about the applicability of the method. To have a valuable article, it is very important to underline how this information can help other future researches.

Response 6: I have introduced the applicability of SS treatment in industry (Line247-249). Therefore, I suggest the further researches focus on the technological parameter of this method in industry.

Thank you for your comments on our paper and we will try to improve it continuously.

Reviewer 4 Report

The manuscript deals on the mitigation of deoxynivalenol contents in wheat, which is a very common contaminated matrix. Different thermal processes have been published regarding the reduction of mycotoxin levels in diverse food commodities, also superheated steam. Some of them are cited in the manuscript, however, in my opinion the obtained results could be compared to other studies performed by SS and also by other heat treatments.

The article is acceptable in the present form; however, some minor remarks listed below should be taken in consideration:

In line 25 I would say common instead of vital.

Lines 42-43: The authors say that many studies have been reported regarding DON stability under high temperatures, however, only one study is cited (number 21).

Lines 43-44: In the sentence “Therefore, SS may be an available way to reduce the content of DON in scabbed wheat with superheated steam” superheated steam is repeated.

In lines 46 and 48 superheated steam can be written as abbreviated form.

Lines 54-55: why these parameters were selected? It was based on other studies?

The first figure is number 2, please revise figure numbers.

Lines 86-87: “Because of the harmful effect of SS on wheat quality, the processing time should be chosen according to the changes of DON content and wheat quality”. How the processing time is chosen depending on DON contents and wheat quality? This sentence is not clear.

Line 90: Did the authors consider that this fact could be related to the initial DON content in sample? When describe the results, that are always referred to sample L but results obtained in sample H are not well described in the text although are showed in the figures. In addition, why the authors consider that 3.8 mg/kg is higher level and 2.3 mg/kg lower level?

Lines 105-106: “In most experiments, the reduction rate of DON in wheat kernels was higher than that of wheat flour”. Are the authors talking about their experiments or other studies? Which studies? Please cite these studies.

Line 134: Why sample L? What about sample H?

Line 205: What does “development time” mean in this context?

Line 206: I think the authors want to say affected instead of effected.

Line 224: The SS process should be described in detail indicating what happens to the sample in each step indicated in the figure. In my opinion this is very useful to understand the reduction process in samples.

Line 233: It is very important to describe the conditions of the cooling and storing steps as they can affect mycotoxin contents in the matrix.

Line 286: in my opinion not only the texture should be evaluated, also the flavour and other characteristics related to the acceptance for consumers. I think the authors must evaluate also these parameters.

Author contributions and funding should be specified.

Author Response

Dear Reviewer:

Thank you very much for your suggestion and comments on our manuscript. Now I’m glad to reply to your comments point by point.

Point 1: The manuscript deals on the mitigation of deoxynivalenol contents in wheat, which is a very common contaminated matrix. Different thermal processes have been published regarding the reduction of mycotoxin levels in diverse food commodities, also superheated steam. Some of them are cited in the manuscript, however, in my opinion the obtained results could be compared to other studies performed by SS and also by other heat treatments.

Response 1: I have compared our results on other studies performed by SS and also by other heat treatments in the revised paper (Line128-139).

The article is acceptable in the present form; however, some minor remarks listed below should be taken in consideration:

Point 2: In line 25 I would say common instead of vital.

Response 2: I have replaced the word “vital” by “common” (Line 33).

Point 3: Lines 42-43: The authors say that many studies have been reported regarding DON stability under high temperatures, however, only one study is cited (number 21).

Response 3: Many articles have concerned on the stability of DON under high temperature. So I have cited 3 more articles that are representative (Line 53, 436-442).

Point 4: Lines 43-44: In the sentence “Therefore, SS may be an available way to reduce the content of DON in scabbed wheat with superheated steam” superheated steam is repeated.

Response 4: I’m sorry that I have made a stupid mistake and I have corrected it in the revised paper.

Point 5: In lines 46 and 48 superheated steam can be written as abbreviated form.

Response 5: The word “superheated steam” has been written as the abbreviated form “SS” (Line 58-61).

Point 6: Lines 54-55: why these parameters were selected? It was based on other studies?

The first figure is number 2, please revise figure numbers.

Response 6: These parameters were selected based on the preliminary experiment and the experiment design principle.

Figure number: All the figure numbers have been corrected in the revised paper.

Point 7: Lines 86-87: “Because of the harmful effect of SS on wheat quality, the processing time should be chosen according to the changes of DON content and wheat quality”. How the processing time is chosen depending on DON contents and wheat quality? This sentence is not clear.

Response 7: That is to say, based on the safe principle (the DON concentration is reduced to the safe level), we should try to finish the processing of wheat with superheated steam in less time. Similarly, we will try to use lower temperature and steam velocity on condition that safety is insured. That is because that when processing time is very long, steam temperature is quite high or steam velocity is not enough is not fit, wheat quality may be affected negatively. The above reason has been added in the revised paper.

Point 8: Line 90: Did the authors consider that this fact could be related to the initial DON content in sample? When describe the results, that are always referred to sample L but results obtained in sample H are not well described in the text although are showed in the figures. In addition, why the authors consider that 3.8 mg/kg is higher level and 2.3 mg/kg lower level?

Response 8: I’m sorry that I haven’t paid attention to this problem. In fact, I can’t confirm whether this fact could be related to the initial DON content in sample. I just listed a result here. Since I can’t explain this phenomenon clearly based on the existing data, I have deleted this result in the revised paper. In the future research, we will try to explain it. Besides, since the DON concentration in Sample L and Sample H have the similar variation trends, so I mainly described Sample L in detail as a representative. Sample H and Sample L are the two samples we get from the farmlands in China and we think that the DON concentrations in the two samples can represent the current situation of DON contamination in the world (please see Reference 2, 3, 4, and 5 at your convenience).

Point 9: Lines 105-106: “In most experiments, the reduction rate of DON in wheat kernels was higher than that of wheat flour”. Are the authors talking about their experiments or other studies? Which studies? Please cite these studies.

Response 9: This conclusion was achieved from other data of our research. Since the data was not listed in this article, I have deleted this sentence.

Point 10: Line 134: Why sample L? What about sample H?

Response 10: Compared to Sample L, the quality of Sample H is worse and DON concentration in Sample H is higher after processed with superheated steam at same conditions. As a result, Sample L is more suitable to be used as biscuit material. Besides, there are similar trends of quality changes between Sample H and Sample L. Therefore, we have used Sample L as the representative sample.

Point 11: Line 205: What does “development time” mean in this context?

Response 11: The word “development time” has appeared in Table 1. In this context, the meaning of “development time” is the same with than in the table. “Development time” means the time when the torque reaches the highest.

Point 12: Line 206: I think the authors want to say affected instead of effected.

Response 12: The word “effected” has been replaced by the word “affected” in the revised paper (Line 244).

Point 13: Line 224: The SS process should be described in detail indicating what happens to the sample in each step indicated in the figure. In my opinion this is very useful to understand the reduction process in samples.

Response 13: Thank you for your great suggestions and the SS process has been described in detail in the revised paper (Line 266-282).

Point 14: Line 233: It is very important to describe the conditions of the cooling and storing steps as they can affect mycotoxin contents in the matrix.

Response 14: The cooling and storing steps has been added in the revised paper (Line 278-282).

Point 15: Line 286: in my opinion not only the texture should be evaluated, also the flavour and other characteristics related to the acceptance for consumers. I think the authors must evaluate also these parameters.

Response 15: Thank you for your great suggestion. We agree with you that flavor is very important. But, what we focus in this paper is the reduction of DON, the rheological properties of dough and the texture of biscuits. Besides, we will not ignore your advice and will do research into the flavor of biscuits in the future carefully.

Point 16: Author contributions and funding should be specified.

Response 16: The above information has been added in the revised paper.

Thank you very much for your suggestions and comments on our manuscript and we will try to improve it continuously.

Round 2

Reviewer 1 Report

Dear authors, thank you for you reply. However, I still feel that this is not a novel and original method. The use of HPLC for DON determination; only DON detemination without its metabolites are my concern because the modified and emerging toxins are one of the reason why mycotoxins are still so popular. In my opinio, in order to involve novelty, you should have at least take one modified DON derivate (3, or 15 AcDON or its glucoside). Thus, I have to submitt  my recommendation as reject.

Author Response

Dear Reviewer:

Thank you very much for your suggestion and comments on our manuscript. Now I’m glad to reply to your comments point by point.

Reviewer comments:

Dear authors, thank you for you reply. However, I still feel that this is not a novel and original method. The use of HPLC for DON determination; only DON detemination without its metabolites are my concern because the modified and emerging toxins are one of the reason why mycotoxins are still so popular. In my opinio, in order to involve novelty, you should have at least take one modified DON derivate (3, or 15 AcDON or its glucoside). Thus, I have to submit my recommendation as reject.

Response: Dear Reviewer, thank you very much for your valuable suggestion about my manuscript. However, it is such an important and big experiment that we can’t finish it in a short time. Therefore, we would like to explain the problem to you based on two articles published in Food Chemistry (Wu et al.) and Food Control (Zhang et al.). According to the two articles, we found that the content of 3-AcDON, 15-AcDON and DON-3-glucoside were all reduced during the making of steamed bread. Therefore, we speculated that DON could not be translated into the above products. The inference still needed verification and we will do research into it in the near future.

Than you again for your valuable suggestion and we will improve our manuscript continuously.

References:

Wu L , Wang B . Transformation of deoxynivalenol and its acetylated derivatives in Chinese steamed bread making, as affected by pH, yeast, and steaming time[J]. Food Chemistry, 2016, 202:149-155.

Zhang H , Wang B . Fate of deoxynivalenol and deoxynivalenol-3-glucoside during wheat milling and Chinese steamed bread processing[J]. Food Control, 2014, 44:86-91.

Reviewer 2 Report

I believe the manuscript has been significantly improved and now warrants publication in Toxins.

Comments for authors:

Line 58: the terminology “low-gluten wheat flour” is misleading, better use “low-protein wheat flour”

Line 156: typo in ...are not significantly different (P<0.05); for significant (P<0.05) and for non significant (P>0.05)

Line 243: indicate in the manuscript that the wheat type used was soft wheat

Line 282: typo in AAC (AACC is correct)

Line 294: indicate in the manuscript that the SPE column was Bond Elut Mycotoxin SPE column

Line 311: The units for the limit of quantification should be expressed in m/m (i.e. 30 µg/kg or 0.03 µg/g); µg/mL (m/v) is not possible.

Author Response

Dear Reviewer:

Thank you very much for your suggestion and comments on our manuscript. Now I’m glad to reply to your comments point by point.

Point 1: Line 58: the terminology “low-gluten wheat flour” is misleading, better use “low-protein wheat flour”

Response 1: The word “low-gluten” has been replaced by “low protein” in the revised paper (Line 58).

Point 2: Line 156: typo in ...are not significantly different (P<0.05); for significant (P<0.05) and for non significant (P>0.05)

Response 2: I’m sorry that I’ve made a mistake and it has been corrected in the revised paper (Line 85 and 152).

Point 3: Line 243: indicate in the manuscript that the wheat type used was soft wheat

Response 3: I have indicated this in the revised paper (Line 243).

Point 4: Line 282: typo in AAC (AACC is correct)

Response 4: I’m sorry that I’ve made a mistake and it has been corrected in the revised paper (Line 281).

Point 5: Line 294: indicate in the manuscript that the SPE column was Bond Elut Mycotoxin SPE column

Response 5: Bond Elut Mycotoxin has been indicated in the revised paper (Line 291).

Point 6: Line 311: The units for the limit of quantification should be expressed in m/m (i.e. 30 µg/kg or 0.03 µg/g); µg/mL (m/v) is not possible.

Response6: Thank you for your useful reminding. I’ve corrected it in the revised paper. After conversion, the limit of quantification was 0.075 μg/g.

Thank you for your comments on our paper and we will try to improve it continuously.

Yours Sincerely,

Yuanxiao Liu

Reviewer 3 Report

The manuscript was improved. I only have one remark:

For Figure 1-3 and Table 1-3, please mention in the footer of the table/figure what is the significance of the letters in the table/figure. 

I know that it is a common idea from biostatistics to use the letters, but to have a perfect manuscript, this information should be added. 

Author Response

Dear Reviewer:

Thank you very much for your suggestion and comments on our manuscript. Now I’m glad to reply to your comments point by point.

Point 1: For Figure 1-3 and Table 1-3, please mention in the footer of the table/figure what is the significance of the letters in the table/figure.

I know that it is a common idea from biostatistics to use the letters, but to have a perfect manuscript, this information should be added.

Response 1: Thank you for your comments. I think it is very useful for us and the meaning of the letters have been added in the revised paper (Line 85 and 152).

Thank you for your comments on our paper and we will try to improve it continuously.

Yours Sincerely,

Yuanxiao Liu
